# Taylor’s Law in Innovation Processes

**DOI:** 10.3390/e22050573

**Published:** 2020-05-19

**Authors:** Francesca Tria, Irene Crimaldi, Giacomo Aletti, Vito D. P. Servedio

**Affiliations:** 1Physics Department, Sapienza University of Rome, P.le Aldo Moro 5, 00185 Rome, Italy; 2IMT School for Advanced Studies Lucca, Piazza San Ponziano 6, 55100 Lucca, Italy; irene.crimaldi@imtlucca.it; 3ADAMSS Center, Università Degli Studi di Milano, 20133 Milan, Italy; giacomo.aletti@unimi.it; 4Complexity Science Hub Vienna, Josefstädter Strasse 39, A-1080 Vienna, Austria; servedio@csh.ac.at

**Keywords:** innovation dynamics, Taylor’s law, adjacent possible, Poisson–Dirichlet process, Pólya’s urn, triangular urn schemes

## Abstract

Taylor’s law quantifies the scaling properties of the fluctuations of the number of innovations occurring in open systems. Urn-based modeling schemes have already proven to be effective in modeling this complex behaviour. Here, we present analytical estimations of Taylor’s law exponents in such models, by leveraging on their representation in terms of triangular urn models. We also highlight the correspondence of these models with Poisson–Dirichlet processes and demonstrate how a non-trivial Taylor’s law exponent is a kind of universal feature in systems related to human activities. We base this result on the analysis of four collections of data generated by human activity: (i) written language (from a Gutenberg corpus); (ii) an online music website (Last.fm); (iii) Twitter hashtags; (iv) an online collaborative tagging system (Del.icio.us). While Taylor’s law observed in the last two datasets agrees with the plain model predictions, we need to introduce a generalization to fully characterize the behaviour of the first two datasets, where temporal correlations are possibly more relevant. We suggest that Taylor’s law is a fundamental complement to Zipf’s and Heaps’ laws in unveiling the complex dynamical processes underlying the evolution of systems featuring innovation.

## 1. Introduction

The laws of Zipf [1,2,3], Heaps [4,5] and Taylor [6,7], which quantify, respectively, the frequency distribution of elements in a given system, the rate at which new elements enter a given system, and fluctuations in that rate, are recognized as the more general statistical laws characterizing complex systems featuring innovations. As such, they also set minimal requirements for the predictions a given modeling scheme should have to correctly address the fundamental mechanisms driving innovation processes. Zipf’s law, or generalized Zipf’s law predicting a frequency-rank distribution of the form $f\left(R\right)=R^{-\beta }$, with 0<β<+∞ (whereas the strict Zipf’s law refers to β=1) characterizes disparate systems, from cities population to earthquakes amplitudes to the frequency of words in written texts, and different explanations for its emergence have been proposed so far [8,9,10]. While Zipf’s law is a static property of the system, Heaps’ law explicitly refers to its evolution and states that the number of distinct elements D(n) when the system consists of *n* elements follows a power law $D\left(n\right)\propto n^{\gamma }$, 0<γ≤1. This points to two fundamental properties shared by different systems related to human activity, from natural language, to the way humans listen to music or interact in a collaborative online systems, or build up collaborations in a research activity: (i) new elements continuously enter the system; (ii) the rate at which innovation occurs slows down with the intrinsic time of the system (when the strict inequality γ<1 holds), e.g., it is easier and easier to continue with established collaborations than linking to new ones. These two simple laws puzzled the scientific community to a large extent, and although it was recognized that in some conditions one law implies the other [11], a general model able to account for both in a common ground, without deriving one from the other, and from microscopic mechanisms, was only recently proposed [12]. This model was based on the notion of the adjacent possible [13], and was generalized in different forms [14,15] to account for higher-level properties of the systems under study.

Taylor’s law was more recently related to the onset of complex behaviour. The law was originally formulated in the context of population ecology, where the observation was made [6] that the variance $\sigma^{2}$ of the population density of different species scales as a power-law of the mean population density: $\sigma^{2}\propto \mu^{b}$, with the exponent 1≲b≲3. While b=1 arises in the case of random distribution of species in the environment, a value b>1 points to correlated patterns. During the past half-century, Taylor’s power law was then observed both in biological and non-biological contexts, from ecology to life-sciences, from physics to economics [7]. In the framework of populations ecology, different mechanisms have been proposed to account for the observed Taylor’s law exponents, from (negative) interaction between species in an ecosystem [16], to correlations between individuals reproduction ability [17]. In [18,19], a simple stochastic multiplicative process is shown to produce Taylor’s exponents ranging from zero to infinity depending on the population’s growth regime, and in this context the value b≃2 observed in real systems is shown to be ascribable to a sampling artifact [20]. In the contest of complex systems, Taylor’s law was measured for the first time in [21] referred to the number of different elements D(n) at a given text’s length *n*. In particular, it was observed that the standard deviation σ[D(n)] as a function of the mean μ[D(n)] scales in written texts as $\sigma \left[D\left(n\right)\right]\propto \mu {\left[D\left(n\right)\right]}^{\beta }$, with β∼1. Here, an exponent β=1/2 was expected if successive introductions of novel elements in the words sequence were independent of previous innovations. This is the case for instance in Simon-like models [22,23], both with constant and time-dependent innovation rate, where the introduction of novelties follows a Poissonian process, respectively homogeneous and inhomogeneous, thus the variance being proportional to the mean. Correlations between the introduction of different novelties have to be considered to predict β>1/2. The model introduced in [12] was recognized [24] to predict exponents for Taylor’s law ranging from 1/2 to 1, depending on the relative importance of the processes of innovation versus reinforcement of old elements, thus better accounting for the values observed in real systems.

The contribution of the present study is twofold. Firstly, we fully characterize the prediction for Taylor’s law of the recently introduced modeling scheme [12] based on the adjacent possible. We recall results for its exchangeable [25,26], counterpart and, relying on known results on triangular urn models, extend them for all the spectrum of the parameter values of the original model, including the non exchangeable region. We further give results for two generalizations of the model, allowing to also predict exponents for Taylor’s law greater than one, as observed in real systems: (i) a version with random quenched parameters and (ii) a version where semantic triggering is introduced, as in [12]. We devote a particular emphasis to the connection of the urn model with triggering with the two parameters Poisson–Dirichlet process [27,28]. The two parameters Poisson–Dirichlet process is the state-of-the-art reference process for language modeling [29], and in its hierarchical form [30], for the search of underlying semantic categories or topics [31], since it reproduces the correct basic statistics of words, namely the Zipf’s and Heaps’ laws and Taylor’s law with exponent β=1. In general, by choosing the underlying stochastic process, that defines the space of probability over which one performs the optimization, we steer the prediction on key statistical features of the system. In this perspective, adopting the best model becomes crucial, and the urn model with triggering opens the way for generalizations that go beyond exchangeability, for instance by considering semantic triggering, as already introduced in [12], thus posing the ground for more effective inference schemes.

Secondly, we extend the observation of Taylor’s law, referred to fluctuations in innovation rate, in several datasets, showing that a non-trivial behaviour of Taylor’s law seems to be universal in those systems. In particular, we consider four datasets related to human activities: (i) written language from a subset of the Gutenberg corpus of English texts; (ii) Twitter hashtags; (iii) a collaborative tagging system (Delicious); (vi) the list of temporarily ordered songs listened by many users in the Last.fm website. We observe how Taylor’s law of the actual sequences of events follows in all the dataset the form $\sigma \left[D\left(n\right)\right]\propto \mu^{\beta }\left[D\left(n\right)\right]$, with β≳1. Furthermore, we highlight how the randomized sequences, obtained by retaining all the elements of the original sequences and changing their temporal order (see Section 5 for details), show different behavior in the Gutenberg corpus and Last.fm compared to Twitter and Delicious. This issue remains still not fully explained, and leaves open the need of a deeper understanding of the process responsible for this behavior.

The paper is organized as follows. In the next section, we recall the urn model with triggering, devoting a particular emphasis to its connection with the two parameters Poisson–Dirichlet process [27,28]. We recall, in particular, how the urn model with triggering can be recast to be equivalent to the latter stochastic process. At the same time, it extends the Poisson–Dirichlet process in the region where the latter is not defined, i.e., in the region of linear innovation growth. In Section 3, relying on known results on triangular urn models [32,33], we characterize the limit distribution for the number of distinct elements D(n) and Taylor’s law for the urn model with triggering (Section 4). In Section 5, we discuss the Taylor’s law in the four datasets mentioned above. Finally, in Section 6, we discuss two different mechanisms that can increase the exponent of Taylor’s law at a value β>1, as observed in the considered real-world systems.

## 2. The Urn Model with Triggering

The urn model with triggering, introduced in [12], is a minimal model based on Pólya’s urn able to reproduce the main statistical signatures of innovation processes, namely Zipf’s, Heaps’ and Taylor’s law. It casts in a mathematical framework the idea of the expansion into the adjacent possible [13,34,35] where the space of possibilities is continuously enlarged, due to the realization of part of them. A crucial element is thus correlations between the emergence of novel elements in the system. The model works as follows. An urn initially contains $N_{0}>0$ distinct balls of different colors. Then, at each time step *t*, a ball is drawn at random from the urn to construct a sequence S of events, and it is put back in the urn. Further,
if the color of the extracted ball is a *new* one, (it appears for the first time in S, i.e., it is a realization of a novelty), then we add $\tilde{\rho }$ balls of the same color plus ν+1 distinct balls of different *new* colors, which were not yet present in the urn; note that we use here the word *new* in two different acceptations: on one hand we refer to events that occur for the first time, on the other one to new colors that enter the space S of eventsif the color of the extracted ball is already present in S, we add ρ balls of the same color.
Therefore, if $C_{t+1}$ is the color of the extracted ball at time t+1 and $D_{t}$ is the number of different colors extracted until time *t*, we have:(1)$$b_{t}:=P\left(C_{t+1}=new\;|\;C_{1},\cdots ,C_{t}\right)=\frac{N_{0}+\nu D_{t}}{N_{0}+\rho t+aD_{t}},$$
where $a:=-\rho +\tilde{\rho }+\nu +1$. Moreover, if *c* denotes an *old* color, we have
(2)$$p_{c,t}:=P\left(C_{t+1}=c\;|\;C_{1},\cdots ,C_{t}\right)=\frac{1+\tilde{\rho }+\rho \left(K_{c,t}-1\right)}{N_{0}+\rho t+aD_{t}}=\frac{\rho K_{c,t}+a-\nu }{N_{0}+\rho t+aD_{t}},$$
where $K_{c,t}$ denotes the number of extractions of the color *c* until time *t*.

### Values of the Model Parameters

Note that we have $p_{c,t}>0$ for each *t* when $\tilde{\rho }>-1$. The model can be defined also for $\tilde{\rho }=-1$, but this implies $b_{t}=1$ and $D_{t}=t$ for all *t*. Moreover, the value ρ=0 is possible, but in that case $p_{c,t}$ would not depend on $K_{c,t}$, e.g., no reinforcement effect would be present. Therefore, we focus on the case $\tilde{\rho }>-1$, ν≥0 and ρ>0.

In [24], an interesting particular case was highlighted. When a=0, i.e., $\tilde{\rho }=\rho -\left(\nu +1\right)$ the above model corresponds to the Poisson–Dirichlet (PD) process, also called Chinese restaurant model [27,28,36]. Indeed, we have
(3)$$b_{t}=\frac{N_{0}+\nu D_{t}}{N_{0}+\rho t}\;\mathrm{and}\;p_{c,t}=\frac{\rho K_{c,t}-\nu }{N_{0}+\rho t}.$$

More precisely, it corresponds to the PD process with parameters α=ν/ρ and $\theta =N_{0}/\rho$. Note that the condition $\tilde{\rho }>-1$ becomes ρ>ν (hence ρ>0) and so 0≤α<1. The particular case ν=0 is also known as Dirichlet process with parameter $\theta =N_{0}/\rho$. Moreover, if we also set ρ=1, we find a particular Dirichlet process, known as the Hoppe’s model [37]. The PD process is a well-known example of exchangeable “species sampling sequence” [27] and a generalization of this process with random weights can be found in [38].

## 3. Triangular Urn Schemes and Innovation Rate

Concerning the behavior of the number of distinct elements $D_{t}$, the above urn model can be seen as a triangular two-color urn scheme [32,33,39,40]. More precisely, we can consider an urn model with the following dynamics. The urn initially contains $N_{0}>0$ black balls. Then, at each time step *t*, a ball is drawn at random from the urn and

if the color of the extracted ball is black, then we replace the extracted ball with a white ball and we add $\tilde{\rho }$ white balls plus ν+1 black balls;if the color of the extracted ball is white, we return the extracted ball in the urn together with ρ additional white balls.

Therefore, in this urn scheme the extraction of a black (resp. white) ball corresponds to the extraction of a *new* (resp. *old*) color in the urn model with triggering. If we denote by $B_{t}$ and $W_{t}$, respectively, the number of black and white balls in the urn at time step *t* and by $\delta_{t}$ a random variable taking values in {0,1} such that $\delta_{t}=1$ if the extracted ball at time step *t* is black, then we have $B_{0}=N_{0}>0$, $W_{0}=0$ and, for each t≥0,
(4)$$\left(\begin{array}{c} B_{t+1}\\ W_{t+1} \end{array}\right)=\left(\begin{array}{c} B_{t}\\ W_{t} \end{array}\right)+\left(\begin{array}{cc} \nu & 0\\ \hat{\rho } & \rho \end{array}\right)\left(\begin{array}{c} \delta_{t+1}\\ 1-\delta_{t+1} \end{array}\right),$$
with $\hat{\rho }:=\tilde{\rho }+1$ and $P\left(\delta_{t+1}=1|\delta_{1},\cdots ,\delta_{t}\right)=B_{t}/\left(B_{t}+W_{t}\right)$. A dynamics of this kind is a two-color urn model with triangular replacement matrix
(5)$$R=\left(\begin{array}{cc} \nu & \hat{\rho }\\ 0 & \rho \end{array}\right).$$

The *balance condition*, which requires that the number of added balls is the same at each time step, independently of the color of the extracted ball, corresponds to the particular case $a=\nu +\hat{\rho }-\rho =0$. Recalling that we are assuming ν≥0, ρ>0 and $\hat{\rho }>0$, the balance condition is possible only if ρ>ν. According to the above notation, we can write $D_{t}=\sum_{k=1}^{t}\delta_{k}$, $B_{t}=N_{0}+\nu \sum_{k=1}^{t}\delta_{k}=N_{0}+\nu D_{t}$, $W_{t}=\hat{\rho }\sum_{k=1}^{t}\delta_{k}+\rho \sum_{k=1}^{t}\left(1-\delta_{k}\right)=\rho t+\left(\hat{\rho }-\rho \right)D_{t}$, so that $B_{t}+W_{t}=N_{0}+\rho t+aD_{t}$. Therefore, when ν>0, the asymptotic behaviour of $D_{t}$ coincides with the one of $B_{t}/\nu$ and from the results in [32,33,39] (simply translating the results proven in that papers in terms of the considered model) we immediately obtain (in the following $\overset{a.s.}{\to }$ means almost sure convergence and $\overset{d}{\to }$ means convergence in the distribution sense):**(Case 0<ν<ρ)**(6)$$t^{-\nu /\rho }D_{t}\overset{a.s.}{⟶}D,$$
where *D* is a suitable random variable with finite moments. In particular, when a=0, the random variable *D* has probability density function given by
$$f\left(x\right)=cx^{N_{0}/\nu }f_{ML}\left(x\right)\;\mathrm{for}\;x>0,$$
where *c* is a normalizing constant and $f_{ML}$ denotes the probability density function of the Mittag-Leffler distribution with parameter ν/ρ. Hence, for a=0, we have
$$\mu \left[D^{q}\right]=\frac{\Gamma \left(N_{0}/\nu +q\right)\Gamma \left(N_{0}/\rho \right)}{\Gamma \left(N_{0}/\nu \right)\Gamma \left(N_{0}/\rho +q\nu /\rho \right)}.$$**(Case ν=ρ)**(7)$$\frac{\ln(t)}{t}D_{t}\overset{a.s.}{⟶}\frac{\rho }{\hat{\rho }}$$
and
(8)$$\ln\left(t\right)\left(\frac{\ln(t)}{t}D_{t}-\frac{\rho }{\hat{\rho }}-\frac{\rho }{\hat{\rho }}\frac{\ln(\ln(t))}{\ln(t)}\right)\overset{d}{⟶}Z,$$
where *Z* is a suitable random variable.**(Case ν>ρ)**(9)$$t^{-1}D_{t}\overset{a.s.}{⟶}\frac{\left(\nu -\rho \right)}{a}$$
and the second-order behaviour depends on the value of ρ/ν. Precisely, denoting by $N(0,\sigma^{2})$ the normal distribution with mean value equal to zero and variance equal to $\sigma^{2}$, we have:
–for ρ/ν<1/2,
(10)$$\sqrt{t}\left(\frac{D_{t}}{t}-\frac{\left(\nu -\rho \right)}{a}\right)\overset{d}{⟶}N\left(0,\sigma^{2}\right)\;\mathrm{with}\;\sigma^{2}=\frac{\nu \left(\nu -\rho \right)\hat{\rho }}{\left(\nu -2\rho \right)a^{2}};$$–for ρ/ν=1/2,
(11)$$\sqrt{t/\ln(t)}\left(\frac{D_{t}}{t}-\frac{\left(\nu -\rho \right)}{a}\right)\overset{d}{⟶}N\left(0,\sigma^{2}\right)\;\mathrm{with}\;\sigma^{2}=\frac{\rho \hat{\rho }}{{\left(\rho +\hat{\rho }\right)}^{2}};$$–for ρ/ν>1/2,
(12)$$t^{1-\rho /\nu }\left(\frac{D_{t}}{t}-\frac{\left(\nu -\rho \right)}{a}\right)\overset{d}{⟶}-\frac{{\left(\nu -\rho \right)}^{1+\rho /\nu }}{\rho a^{1+\rho /\nu }}V,$$
where *V* is a suitable random variable.

For the degenerate case ν=0, we trivially have $B_{t}=N_{0}$ for each *t*. Moreover, we recall that ρ>0 and $W_{t}-\rho t=\left(\hat{\rho }-\rho \right)D_{t}$. Hence, when $\hat{\rho }\neq \rho$, the asymptotic behaviour of $D_{t}$ follows from the results on $W_{t}$ (see [33]), that is we have
(13)$$\frac{D_{t}}{\ln(t)}\overset{a.s.}{⟶}\frac{N_{0}}{\rho }$$
and
(14)$$\sqrt{\ln(t)}\left(\frac{D_{t}}{\ln(t)}-\frac{N_{0}}{\rho }\right)\overset{d}{⟶}N\left(0,N_{0}/\rho \right).$$

The balance condition with ν=0 means $\hat{\rho }=\rho$ and in this case we have a Dirichlet process with parameter $\theta =N_{0}/\rho$ and the above convergence results still hold true.

In particular, the above convergence results imply $D_{t}\propto t^{\nu /\rho }$ for 0<ν<ρ, $D_{t}\propto t/\ln\left(t\right)$ for ν=ρ, $D_{t}\propto t$ for ν>ρ and, finally, $D_{t}\propto \ln\left(t\right)$ for ν=0 (see also [12,24]).

## 4. Taylor’s Law

The Taylor’s law connects the standard deviation of a random variable to its mean. In the considered model, when the balance condition a=0 is satisfied, we can obtain explicit formulas for the moments of $D_{t}$: indeed, from [33,41], using the relation $B_{t}=N_{0}+\nu D_{t}$, we get for ν>0
$$\begin{array}{cc} \mu [D_{t}] & =\frac{N_{0}\Gamma \left(N_{0}/\rho \right)}{\nu \Gamma (N_{0}/\rho +\nu /\rho )}t^{\nu /\rho }+O\left(t^{\nu /\rho -1}\right)\\ \sigma^{2}\left[D_{t}\right] & =\frac{N_{0}}{\nu^{2}}\left[\left(N_{0}+\nu \right)\frac{\Gamma (N_{0}/\rho )}{\Gamma (N_{0}/\rho +2\nu /\rho )}-N_{0}\frac{\Gamma {\left(N_{0}/\rho \right)}^{2}}{\Gamma {\left(N_{0}/\rho +\nu /\rho \right)}^{2}}\right]t^{2\nu /\rho }+O\left(t^{\nu /\rho }\right)\;. \end{array}$$

Therefore, we find $\sigma \left[D_{t}\right]\propto \mu \left[D_{t}\right]$. For the Dirichlet process (ν=0, a=0), we simply have
(15)$$\begin{array}{cc} \mu [D_{t}] & =\sum_{k=1}^{t}E\left[\delta_{k}\right]=\sum_{k=1}^{t}\frac{N_{0}}{N_{0}+\rho t}∼\frac{N_{0}}{\rho }\ln\left(t\right)\;\mathrm{and}\\ \sigma^{2}\left[D_{t}\right] & =\sum_{k=1}^{t}\sigma^{2}\left[\delta_{k}\right]=\sum_{k=1}^{t}\frac{N_{0}\rho t}{{\left(N_{0}+\rho t\right)}^{2}}∼\frac{N_{0}}{\rho }\ln\left(t\right)\;, \end{array}$$
and so $\sigma \left[D_{t}\right]\propto \mu {\left[D_{t}\right]}^{\frac{1}{2}}$.

To our knowledge, in the unbalanced case we have not explicit formulas for the first and the second asymptotic moments of $D_{t}$. Here, we conjecture that suitable uniform integrability conditions hold for the convergence results in Section 3 in order to infer the convergence of the first two moments having only almost sure convergence and convergence in distribution (see, e.g., [42,43]). In other words, we leverage the convergence results in Section 3 in order to guess the corresponding Taylor’s law.

**(Case 0<ν<ρ)** From the almost sure convergence (Equation 6), we guess $\sigma \left[D_{t}\right]\propto \mu \left[D_{t}\right]$, where the constant of proportionality is σ[D]/μ[D].**(Case ν=ρ)** Since the limit in (Equation 7) is a constant, we can not exploit the almost sure convergence (Equation 7) in order to obtain a Taylor’s law as done for the previous case 0<ν<ρ. However, from the convergence in distribution (Equation 8), we can guess
$$\frac{\ln(t)}{t}\mu \left[D_{t}\right]⟶\frac{\rho }{\hat{\rho }}$$
and
$$\begin{array}{cc} \frac{\ln{\left(t\right)}^{4}}{t^{2}}\sigma^{2}\left[D_{t}\right] & =\ln{\left(t\right)}^{2}\sigma^{2}\left[\frac{\ln(t)}{t}D_{t}-\frac{\rho }{\hat{\rho }}\right]\\ \; & =\mu \left[\ln{\left(t\right)}^{2}{\left(\frac{\ln(t)}{t}D_{t}-\frac{\rho }{\hat{\rho }}\right)}^{2}\right]-{\left(\mu \left[\ln\left(t\right)\left(\frac{\ln(t)}{t}D_{t}-\frac{\rho }{\hat{\rho }}\right)\right]\right)}^{2}⟶\sigma^{2}\left[Z\right]. \end{array}$$Hence, combining together the above two limit relations, we find
$$\sigma^{2}\left[D_{t}\right]∼\sigma^{2}\left[Z\right]\frac{{\hat{\rho }}^{2}}{\rho^{2}}\;\frac{\mu {\left[D_{t}\right]}^{2}}{\ln{\left(t\right)}^{2}},\;\mathrm{that}\;\mathrm{is}\;\sigma \left[D_{t}\right]\propto \frac{\mu [D_{t}]}{\ln(t)}∼\frac{\mu [D_{t}]}{\ln\left(\mu \left[D_{t}\right]\right)+\ln\left(\hat{\rho }/\rho \right)}.$$**(Case ν>ρ)** Since $D_{t}/t\in \left[0,1\right]$ for all *t*, the almost sure convergence (Equation 9) implies the convergence of the moments (see [43]) for that equation. However, it is not enough in order to get a Taylor’s law, but we need to use (Equation 10), (Equation 11) and (Equation 12). First of all, we observe that
$$t^{-2}\sigma^{2}\left[D_{t}\right]=\sigma^{2}\left[\frac{D_{t}}{t}-\frac{\left(\nu -\rho \right)}{a}\right]=\mu \left[{\left(\frac{D_{t}}{t}-\frac{\left(\nu -\rho \right)}{a}\right)}^{2}\right]-{\left(\mu \left[\frac{D_{t}}{t}-\frac{\left(\nu -\rho \right)}{a}\right]\right)}^{2}.$$Hence:–for ρ/ν<1/2, we guess from (Equation 10) that the first term on the right hand of the above equality behaves as $\sigma^{2}/t$, while the second term is o(1/t), and so we get $\sigma^{2}\left[D_{t}\right]∼\sigma^{2}t$ and
$$\sigma \left[D_{t}\right]\propto \mu {\left[D_{t}\right]}^{\frac{1}{2}}$$
with the constant of proportionality equal to $\sigma \sqrt{a/(\nu -\rho )}$;–for ρ/ν=1/2, we guess from (Equation 11) that the first term on the right hand of the above equality behaves as $\sigma^{2}\ln\left(t\right)/t$, while the second term is o(ln(t)/t) and so we get $\sigma^{2}\left[D_{t}\right]∼\sigma^{2}t\ln\left(t\right)$ and
$$\sigma \left[D_{t}\right]\propto \mu {\left[D_{t}\right]}^{\frac{1}{2}}{\left(\ln\left(\mu \left[D_{t}\right]\right)+\ln\left(a/\rho \right)\right)}^{\frac{1}{2}}$$
with the constant of proportionality equal to $\sigma \sqrt{a/\rho }$;–for ρ/ν>1/2, we guess from (Equation 12) that the first term and the second term on the right hand of the above equality behave as $\mu \left[Z^{2}\right]t^{2(\rho /\nu -1)}$ and $\mu {\left[Z\right]}^{2}t^{2(\rho /\nu -1)}$ respectively and so we get $\sigma^{2}\left[D_{t}\right]∼\sigma^{2}\left[Z\right]t^{2\rho /\nu }$ and
$$\sigma \left[D_{t}\right]\propto \mu {\left[D_{t}\right]}^{\rho /\nu }$$
with the constant of proportionality equal to $\sigma \left[Z\right]{\left(a/(\nu -\rho )\right)}^{\rho /\nu }$.

In the degenerate case ν=0, from the almost sure convergence (Equation 13) we guess $\frac{1}{\ln(t)}\mu \left[D_{t}\right]\to \frac{N_{0}}{\rho }$. Moreover, we observe that
$$\frac{1}{\ln(t)}\sigma^{2}\left[D_{t}\right]=\ln\left(t\right)\sigma^{2}\left[\frac{D_{t}}{\ln(t)}-\frac{N_{0}}{\rho }\right]=\mu \left[\ln\left(t\right){\left(\frac{D_{t}}{\ln(t)}-\frac{N_{0}}{\rho }\right)}^{2}\right]-{\left(\mu \left[\sqrt{\ln(t)}\left(\frac{D_{t}}{\ln(t)}-\frac{N_{0}}{\rho }\right)\right]\right)}^{2}$$
and, from the convergence in distribution (Equation 14), we guess that the first term on the right hand of above equality converges to $N_{0}/\rho$, while the second term converges to zero. Hence, we find $\sigma^{2}\left[D_{t}\right]∼N_{0}\ln\left(t\right)/\rho$ and so $\sigma \left[D_{t}\right]\propto \mu {\left[D_{t}\right]}^{\frac{1}{2}}$.

All the above theoretical predictions are supported by simulations, shown in Figure 1, left. We also report in Figure 1, right, the Taylor’s law for the corresponding reshuffled sequences, where the elements are the same as in the original sequences but the temporal order (their ordering in the sequence) is lost. For a discussion on the shuffling procedure see the next section.

## 5. Taylor’s Law in Real World Systems

We base our empirical analysis on four datasets whose content is the result of voluntary human activity. The first dataset consists of english written texts from the on-line collection of public domain books hosted at the Gutenberg project [44]. This dataset was crawled in year 2007. From that, we selected the longest 100 books. In this case, innovations are represented by new words entering the text. The second dataset contains the list of songs listened by 1000 Last.fm users until the 5th of May 2009 [45]. This list has been ordered according to the time of listening. Songs listened for the first time in the Last.fm platform are considered as innovations. The third dataset contains the time ordered list of tags in the platform Del.icio.us [46], where users used keywords (tags) to categorize bookmarked URLs. The dataset contains the tag sequence of users activity from early 2004 up to the end of 2006. The Del.icio.us platform has been discontinued. We treat as innovation the very first usage of a tag in Del.icio.us. The fourth and last dataset contains the time ordered sequence of the 10% of all the hashtags appeared in January 2013 on the micro-blogging platform Twitter [47]. Also in this database, we consider brand new hashtags entering the system as innovations. All these four datasets were already studied in previous works [12,14].

In order to estimate the average number of different tokens and their standard deviation, we preprocessed data such to split them into sequences of given fixed length. We use the generic term *token* to address the elements of the sequences, which are words in the Gutenberg dataset, song titles in Last.fm, tags in Del.icio.us and hashtags in Twitter. In Gutenberg we consider the natural splitting, each sequence being a book. To obtain sequences with the same length, we cut all the books at the length of the shortest one, so that we extracted the first 200,000 words of each of the 100 books. In Del.icio.us we took the last $2\times 10^{7}$ tags and split them into 1000 chunks with 20,000 tags each. In Last.fm, we selected the last $19\times 10^{6}$ titles and split them into 190 chunks of length 100,000. In Twitter we selected the last $346\times 10^{5}$ hashtags and created 346 chunks of length 100,000.

The estimation of the average number of distinct tokens, as well as the standard deviation, is done by determining the number of distinct tokens appeared before a certain position in all the split chunks in parallel. For example, in Gutenberg, we count the number of different words D(N) appeared after *N* total words for all the M=100 books and calculate
(16)$$\mu \left(N\right)=\sum_{i=1}^{M}D_{i}\left(N\right)/M\;\;\;\mathrm{and}\;\;\;\sigma \left(N\right)=\sqrt{\sum_{i=1}^{M}{\left(D_{i}\left(N\right)-\mu \left(N\right)\right)}^{2}/M}\;.$$

We plot these two quantities one against each other for each *N* and display the result in Figure 3.

In order to evaluate the influence of the token macroscopic statistical properties, e.g., the Zipf’s law, on Taylor’s law, we destroy the correlations by reshuffling the sequences. We perform two different shuffles, with increasing randomization, as displayed in Figure 2. In the first one, which we denote as *parallel file random*, we shuffle the tokens inside the same sequence. In the other one, which we call *parallel random*, we shuffle the tokens throughout the all sequences. The results of these randomization schemes on Taylor’s law are shown in Figure 3. Let us first comment that the Del.icio.us and Twitter datasets feature a Taylor exponent, that is the exponent β in the relation $\sigma \left(N\right)\propto \mu {\left(N\right)}^{\beta }$, approximately equal to one. This behavior is well reproduced by the urn model with triggering discussed in Section 2, in the parameters region ν<ρ (refer to Figure 1), that is the region where its exchangeable counterpart, namely the two parameters Poisson–Dirichlet process, is defined. Conversely, the Gutenbeng and Last.fm datasets show a significant deviation from the linear relation between the standard deviation and mean of $D_{t}$, featuring a Taylor’s exponent β>1. The simple urn model with triggering, as well as the two parameters Poisson–Dirichlet process, fails in predicting this deviation from an unitary exponent. However, simple generalization of the considered model can account for it. In the next section we will discuss two possible approaches leading to similar effect on the Taylor’s exponent. Before doing that, we wish to further comment on the results obtained on the reshuffled sequences. The *parallel random* procedure produces asymptotically a trivial (equal to 1/2) Taylor’s exponent for all the datasets, and this reflects the fact that a random sampling from a power law distribution produces a Taylor’s exponent β=1/2 [7,21]. The (*parallel file random*) procedure poses the need to distinguish again the Gutenbeng and Last.fm datasets from Del.icio.us and Twitter. While in the latter datasets the locally reshuffled sequences behave essentially as the original (temporarily ordered) one, in the first two dataset a peculiar behavior of the locally reshuffled sequences emerges, similar in the two datasets and stable against different choices of sets of books in the Gutenberg dataset (Figure 4). The discrepancy between the Taylor’s law in the reshuffled sequences with respect to the original ones points to the fact that randomly sampling from different power law distributions cannot account for the observations, and a different dynamical process has to be considered. The exact mechanism leading to the observed behavior remains an open question, that calls for a more detailed analysis probably adopting hierarchical models, where correlations between the words distributions in different books are taken into account. This will be the topic of a further work.

## 6. Two Mechanisms that Increase Fluctuations

We here propose two mechanisms that generalize the basic model and that are able to account for that higher exponent. On the one hand, increasing fluctuations can be obtained by a quenched stochasticity in the model parameters. That is, each book can be considered as an instance of the considered stochastic process with parameters extracted from a given probability distribution. The term quenched refer to the fact that the parameters are extracted from each realization of the process and remain fixed all along the sequence generation. As a second mechanism, we consider the urn model with semantic triggering introduced in [12] to account for observed clusterization in the emergence on novelties.

### 6.1. Random Parameters

For the sake of analytical simplicity, we here discuss in detail only the case with ν as the random parameter. We show from simulations that similar behaviors are obtained when we take ρ or $N_{0}$ as the random (Figure 5).

**(Case ν>ρ)** As seen before, the Taylor’s exponent in the case ν>ρ is always smaller than 1. Suppose now that ρ and $\hat{\rho }$ are constants and there exists a random variable $X_{0}$, with $\sigma^{2}\left[X_{0}\right]>0$, that gives the value of ν. Given the value of $X_{0}$, the urn process behaves as described before. If $X_{0}$ is concentrated on (ρ,+∞), that is $X_{0}/\rho >1$ almost surely, then, on the event $\left\{X_{0}=\nu \right\}$, the sequence $D_{t}/t$ converges almost surely to the value $\left(\nu -\rho \right)/\left(\nu +\hat{\rho }-\rho \right)$. Therefore, since $D_{t}/t$ is bounded, we have [43]
$$\mu \left[D_{t}\right]∼t\mu \left[\frac{\left(X_{0}-\rho \right)}{\left(X_{0}+\hat{\rho }-\rho \right)}\right]\;\mathrm{and}\;\mu \left[D_{t}^{2}\right]∼t^{2}\mu \left[\frac{{\left(X_{0}-\rho \right)}^{2}}{{\left(X_{0}+\hat{\rho }-\rho \right)}^{2}}\right].$$Therefore, by setting $\tilde{D}=\frac{\left(X_{0}-\rho \right)}{\left(X_{0}+\hat{\rho }-\rho \right)}=1-\frac{\hat{\rho }}{X_{0}+\hat{\rho }-\rho }$, we find
$$\sigma^{2}\left[D_{t}\right]∼\frac{\sigma^{2}\left[\tilde{D}\right]}{\mu {\left[\tilde{D}\right]}^{2}}\mu {\left[D_{t}\right]}^{2},\;\mathrm{that}\;\mathrm{is}\;\sigma \left[D_{t}\right]\propto \mu \left[D_{t}\right].$$This means that while a deterministic parameter ν>ρ gives a Taylor’s exponent smaller than 1, a random parameter ν, with ν/ρ>1 almost surely, gives a Taylor’s exponent equal to 1.**(Case ν<ρ)** As seen before, the Taylor’s exponent in the case ν<ρ is equal to 1. Suppose now, as before, that $X_{0}$ is a random variable, with $\sigma^{2}\left[X_{0}\right]>0$, that gives the value of ν, while the other parameters are constant. If $X_{0}$ is concentrated on (0,ρ), that is $X_{0}/\rho <1$ almost surely, then, on the event $\left\{X_{0}=\nu \right\}$, the sequence $t^{-\nu /\rho }D_{t}$ converges almost surely to a suitable random variable $D_{\nu }$. Moreover, from [33], we have
(17)$$\begin{array}{cc} \; & g_{q}\left(\nu \right):=\mu \left[D_{\nu }^{q}\right]=\frac{\Gamma (N_{0}/\nu +q)}{\Gamma \left(N_{0}/\nu \right)\Gamma \left(q\nu /\rho \right)}\int_{0}^{+\infty }x^{q\nu /\rho -1}{\left(1+\frac{\nu }{\rho }x^{\nu /\rho }h\left(x\right)\right)}^{-N_{0}/\nu -q}\;dx\\ \; & \mathrm{with}\\ \; & h\left(x\right)=\int_{0}^{x}\left[1-{\left(1+u\right)}^{-\hat{\rho }/\rho }\right]u^{-\nu /\rho -1}\;du. \end{array}$$Assuming, as in the previous section, a condition of uniform integrability, we can say that
$$\mu \left[D_{t}\right]∼\mu \left[t^{X_{0}/\rho }g_{1}\left(X_{0}\right)\right],$$
where $g_{1}\left(\nu \right)$ is the function given in (Equation 17) with q=1. Similarly,
$$\mu \left[D_{t}^{2}\right]∼\mu \left[t^{2X_{0}/\rho }g_{2}\left(X_{0}\right)\right],$$
where $g_{2}\left(\nu \right)$ is the function given in (Equation 17) with q=2.If we neglect the terms $g_{q}\left(X_{0}\right)$ in the above mean values, we have
$$\begin{array}{cc} \mu [D_{t}] & ∼\mu \left[t^{X_{0}/\rho }\right]=\mu \left[e^{X_{0}\ln\left(t\right)/\rho }\right]=G_{X_{0}}\left(\ln\left(t\right)/\rho \right),\\ \mu [D_{t}^{2}] & ∼\mu \left[t^{2X_{0}/\rho }\right]=\mu \left[e^{2X_{0}\ln\left(t\right)/\rho }\right]=G_{X_{0}}\left(2\ln\left(t\right)/\rho \right), \end{array}$$
where $G_{X_{0}}$ is the moment-generating function of $X_{0}$. For instance, if $X_{0}$ is uniformly distributed on (0,ρ), we get
$$\mu \left[D_{t}\right]∼\frac{t-1}{\ln(t)}∼\frac{t}{\ln(t)},\;\mu \left[D_{t}^{2}\right]∼\frac{t^{2}-1}{2\ln(t)}∼\frac{t^{2}}{2\ln(t)},$$
and so $\sigma^{2}\left[D_{t}\right]\propto \mu {\left[D_{t}\right]}^{2}\ln\left(t\right)$. Finally, if we use the approximation $\ln\left(\mu \left[D_{t}\right]\right)\propto \ln\left(t\right)$, we obtain $\sigma \left[D_{t}\right]\propto \mu \left[D_{t}\right]{\left(\ln\left(\mu \left[D_{t}\right]\right)\right)}^{\frac{1}{2}}$.Similarly, if $X_{0}$ is exponentially distributed on (0,ρ), that is $f_{X_{0}}\left(x\right)=c\left(\rho ,\lambda \right)e^{-\lambda x}I_{\left(0,\rho \right)}\left(x\right)$ with λ>0 and $c\left(\rho ,\lambda \right)=\lambda /\left(1-e^{-\rho \lambda }\right)$, we get
$$\begin{array}{cc} \mu [D_{t}] & ∼c\left(\rho ,\lambda \right)\rho e^{-\rho \lambda }\frac{\left(t-e^{\rho \lambda }\right)}{\left(\ln(t)-\rho \lambda \right)}∼c\left(\rho ,\lambda \right)\rho e^{-\rho \lambda }\frac{t}{\ln(t)},\\ \mu [D_{t}^{2}] & ∼c\left(\rho ,\lambda \right)\rho e^{-\rho \lambda }\frac{\left(t^{2}-e^{\rho \lambda }\right)}{\left(2\ln(t)-\rho \lambda \right)}∼c\left(\rho ,\lambda \right)\rho e^{-\rho \lambda }\frac{t^{2}}{2\ln(t)}, \end{array}$$
and so, as above, $\sigma \left[D_{t}\right]\propto \mu \left[D_{t}\right]\ln{\left(t\right)}^{\frac{1}{2}}\propto \mu \left[D_{t}\right]{\left(\ln\left(\mu \left[D_{t}\right]\right)\right)}^{\frac{1}{2}}$.From Figure 5 we see that the above predictions are valid asymptotically, after a long transient where a law $\sigma \left[D_{t}\right]\propto \mu {\left[D_{t}\right]}^{\beta }$, β>1 seems to be valid.

### 6.2. Urn Model with Semantic Triggering

For the sake of completeness, we recall here the urn model with semantic triggering introduced in [12], where it was shown that this generalization with respect to the basic model discussed in Section 2 was crucial in order to reproduce higher level features ruling the introduction of novelties in real systems. Let us again consider an urn U initially containing $N_{0}>0$ distinct balls with different colors. Each ball is endowed by a color and by a label as well. Balls with different colors can share the same label, each label defining a semantic group, while balls with different labels necessarily have different colors. The $N_{0}$ balls belong to $N_{0}/\left(\nu +1\right)$ groups, the elements in the same group sharing a common label. In the following, we will say that an element *a* triggered the enter in the urn of the element *b*, if the element *b* is one of the ν+1 elements added in the urn when *a* is drawn for the first time. We thus define the following process. To construct the sequence S, we randomly choose the first element. Then, at each time step *t*:(i)we give weight 1 to: (a) each element in U with the same label, say *C*, as $s_{t-1}$ (the last element added in the sequence), (b) to the element that triggered the enter in the urn of $s_{t-1}$, and (c) to the elements triggered by $s_{t-1}$; a weight η≤1 is given to any other element in U;(ii)The element $s_{t}$ is chosen by drawing randomly from U, each element with a probability proportional to its weight;(iii)the element $s_{t}$ is added to the sequence S and put back into U along with ρ additional copies of it;(iv)if and only if the chosen element $s_{t}$ is new (i.e., it appears for the *first time* in the sequence S), ν+1 brand new distinct elements (balls with different colors, not yet present in the urn), all with a common brand new label, are added to U.

We thus introduced a mechanism through which the occurrence of a ball with a given label facilitates further occurrences, close in time, of other balls with the same label, i.e., semantically related to it. Note that if η=1 the dynamics of this model reduces to that of the model described in Section 2. We do not analyzed in details the behavior of this model (the interested reader can refer to [12], but we remind here that it produces again power laws for the Heaps and Zipf’s laws, with exponents respectively $\min\left(\frac{\nu \eta }{\rho },1\right)\leq \gamma \leq \min\left(\frac{\nu }{\rho },1\right)$ and 1/γ. The behavior for the Taylor’s law is reported in Figure 5 for some choices of the model parameters, showing that it also account for an exponent β>1. For the sake of completeness, we show in Figure 5 also a case where the model with semantic triggering is coupled with a random choice of the model parameters, observing that this does not lead to any substantial different behavior.

## 7. Conclusions

In this paper, we discussed predictions for the Taylor’s law both of a recently introduced modeling scheme based on the notion of the adjacent possible [12], and in four open systems characterized by human activities, where a notion of innovation can be defined. We obtained rigorous mathematical predictions relying on known results for triangular urn models. We supported analytical results and conjectures with simulations of the discussed stochastic process. Further, contrasting model’s predictions and observations from real data, we proposed two, not necessary alternative, generalizations of the model to account for deviations of real data from a pure linear dependence of $\sigma [D_{t}]$ from $\mu [D_{t}]$. Namely, we consider the effect of a quenched stochasticity of the model parameters, and the introduction of semantic correlations, already discussed in [12].

By providing a rigorous mathematical framework to characterize the recently introduced urn model with triggering, the present paper opens the way to a deeper comprehension of the basic mechanisms underlying the observed universalities. On the other hand, a careful analysis of real data highlights relevant observables that unveil distinct behaviours in different systems, possibly due to varying degrees of correlations. A deeper understanding of this subtle behavior could shed some light on distinctive features of human language or on the cognitive and social pressure driving cultural production and fruition.

We finally note that we do not consider here hierarchical models, which we plan to investigate in further works. Hierarchical generalizations of the Poisson–Dirichlet process have proved to be extremely promising in inference problems adopting a Bayesian approach, such as topic modeling in textual corpora. We think that a hierarchical approach can be fundamental to reproduce further statistical features observed in written texts and not already fully explained, such, for instance, the double slope observed in Zipf’s law in large text corpora and the subtle behaviour of Taylor’s law discussed in Section 5.

## Figures and Tables

**Figure 1 entropy-22-00573-f001:**
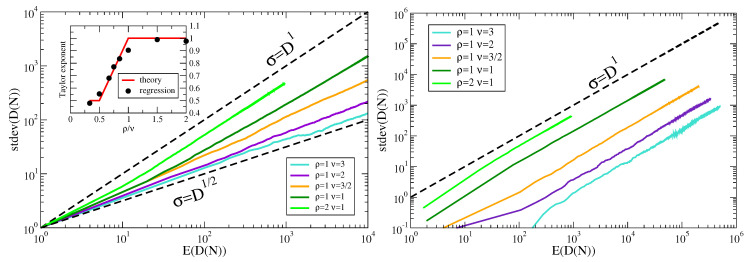
**Taylor’s law in the urn model with triggering.***Left:* Taylor’s law from 100 realizations of the stochastic process described in Section 2 (the urn model with triggering), for each of the indicated values parameters. The values of parameters are chosen in order to have a representative curve for each of the analyzed regimes, i.e., ρ<ν/2, ρ=ν/2, ν/2<ρ<ν, ρ=ν, ρ>ν. Each realization is a sequence of $10^{6}$ elements. *Right:* Taylor’s law from the same sequences as in the left side picture, individually reshuffled so that to loose the temporal order (refer to the *parallel file random* reshuffling procedure discussed in Section 5 and in Figure 2).

**Figure 2 entropy-22-00573-f002:**
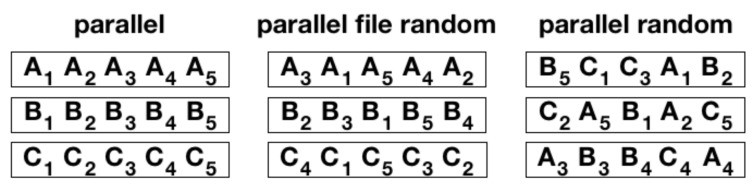
**Shuffling procedures.** In this example we consider three different streams A, B, C, consisting of five tokens each. When the analysis is carried in *parallel* the streams are aligned respecting their natural order (left panel). In the *parallel file random* case (middle panel), each stream is reshuffled singularly. Eventually, the parallel random case shuffles the elements all together (right panel).

**Figure 3 entropy-22-00573-f003:**
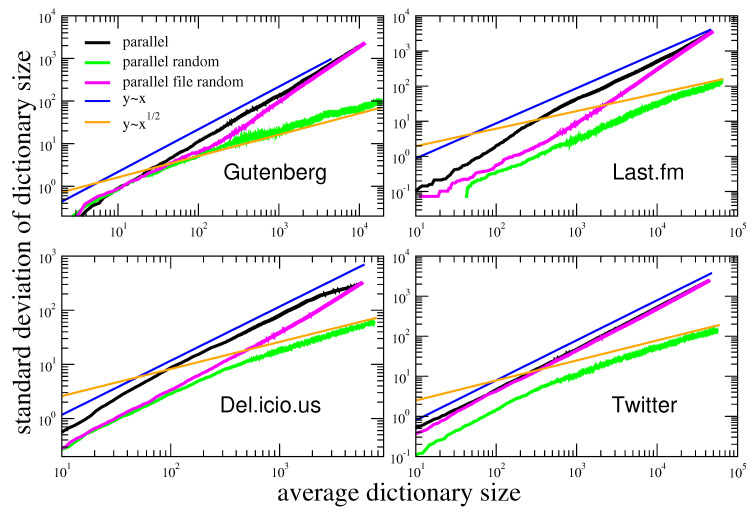
**Taylor’s law in real systems and in their randomized instances.** The standard deviation σ(N) of the number of different tokens after *N* total tokens appeared, is plotted vs the average number of different tokens μ(N) in four different datasets. The shuffled counterparts are also evaluated. The shuffling schemes are shown in Figure 2.

**Figure 4 entropy-22-00573-f004:**
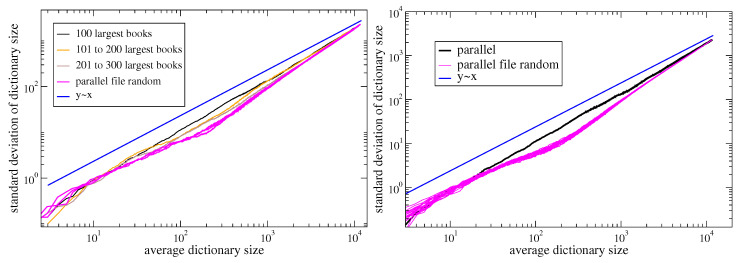
**Stability of Taylor’s law results in the Gutenberg corpus.***Left:* the analogous of Figure 3 (top left) for three different sets of M=100 books from the Gutenberg corpus. *Right*: as in Figure 3 (top left), with 20 different realizations of the *parallel file random* reshuffling procedure. We see that the difference between the curve referred to the ordered sequences and those referred to the reshuffled ones is much higher than fluctuations due to different realizations of the reshuffling.

**Figure 5 entropy-22-00573-f005:**
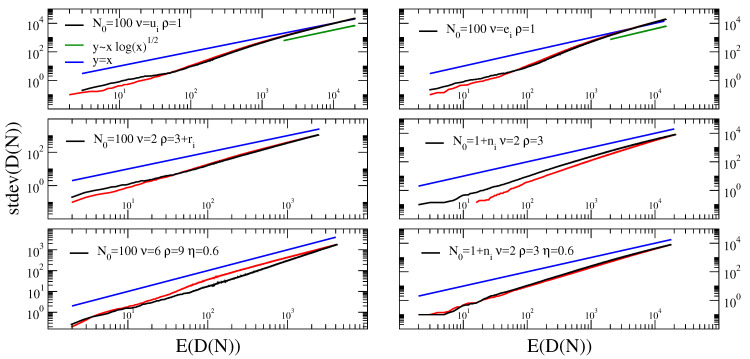
**Taylor’s law in the urn model with triggering and quenched stochasticity of the parameters and in the urn model with semantic triggering.***Top:* Taylor’s law in the urn model with triggering, with parameter’s $N_{0}=100$, ρ=1 and ν is randomly extracted for each simulation of the process from a uniform distribution on the interval (0,1) (left) and from an exponential distribution on the interval (0,1) and parameter λ=1 (right), as discussed in the main text. *Center:* Taylor’s law in the urn model with triggering, with parameter’s respectively: (left) $N_{0}=100$, ν=2, $\rho =3+r_{i}$, with $r_{i}$ randomly extracted for each simulation of the process from an exponential distribution with mean $\bar{r_{i}}=1$; ν=2, ρ=3, $N_{0}=1+n_{i}$, with $n_{i}$ randomly extracted for each simulation of the process from an exponential distribution with mean $\bar{n_{i}}=10^{4}$. *Bottom:* (left) Taylor’s law in the urn model with semantic triggering, with parameters $N_{0}=100$, ν=6, ρ=9, η=0.6; (right) Taylor’s law in the urn model with semantic triggering, with parameters ν=2, ρ=3, η=0.6, $N_{0}=1+n_{i}$, with $n_{i}$ randomly extracted for each simulation of the process from an exponential distribution with mean $\bar{n_{i}}=10^{4}$. The parameters of the simulations were chosen such to lie in the regime ρ<ν. The parameter η=0.6 used in the bottom graphs was chosen in the regime where the Heaps’ and Zipfs’ laws feature exponents compatible with those observed in real systems. In all the figures the curves for the Taylor’s law are constructed from 100 independent realizations of the process (M = 100 in Equation (Equation 16)).

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
