# Peer review of "Taylor’s Law in Innovation Processes"

_entropy, 2020, doi:10.3390/e22050573_

Round 1

Reviewer 1 Report

In this manuscript, the authors study both within a formal framework and using real-world data, the so-called Taylor law, which roughly speaking characterizes the scaling of the amplitude of the fluctuations of the number of innovations as a function of its mean number, in systems involving human activity. The authors wish in particular to understand why in real-world data, the standard deviation of these fluctuations is reported to scale approximately like the mean, while normal fluctuations would rather imply that the standard deviation scales like the square-root of the mean.

The basic framework put forward is that of urn models, where balls of several possible colors are drawn at random from the urn and replaced, together with the addition of new balls of new or already existing color depending on the color of the ball that has been picked up. In this case, innovation consists of introducing new ball colors into the urn. By exploring different versions of the model, the authors are able to recover the anomalous scaling of the standard deviation with the mean. Four different sets of real-world data are analyzed to support this scaling law. However, two out of the four data sets exhibit stronger temporal correlations that require the introduction of a generalized model.

I find the manuscript clear and well-written, presenting the analytics and the data analysis in a consistent and complementary manner. The analytical results are well exposed and relevant to the analyzed data, with as often some unexpected behavior of the data which leads the authors to partly reconsider the model, while keeping the general framework. I can thus support publication of the manuscript.

Author Response

We thank you for your appreciation of our work

Reviewer 2 Report

In my opinion the present paper is a valuable addition to the previous work of (some of the) the authors to the dynamics of innovation and correlated novelties. The paper nicely combines data with a new model and highlights issues should be investigated further. It  is a scientifically honest work, well written and easy to follow through out.  I only suggest to add and comment the few references below: there are very simple models which give Taylor Law with any desired exponent (see references 1 and 2) and also it may happen Taylor Law to be a statistical artifact (reference 3).  Otherwise I recommend  publication as it is.

1. Cohen J. E., Xu M., Schuster W. S. F. Proc. Biol. Sci. 280(1757):20122955 (2013)

2. Cohen J. E. Theor Popul Biol 93:30-37 (2014)

3. Giometto A. et al. PNAS 112 (25), 7755-7760 (2015)

Author Response

We thank you for your appreciation of our work and for your suggestion of relevant references. To include your suggestion we added the following paragraph in the Introduction:

“In the framework of populations ecology, different mechanisms have been proposed to account for the observed Taylor's law exponents, from (negative) interaction between species in an ecosystem~\cite{KILPATRICK2003}, to correlations between  individuals reproduction  ability~\cite{BALLANTYNE2007}. In~\cite{Cohen2013,Cohen2014}, a simple stochastic multiplicative process is shown to produce Taylor's exponents ranging from zero to infinity depending on the population's growth regime, and in this context the value $b \simeq 2$ observed in real systems is shown to be ascribable to a sampling artifact~\cite{Giometto2015}. “

quoting the suggested papers along with the additional papers:

- KILPATRICK, A. M; IVES, A.R. Species interactions can explain Taylor’s power law for ecological time series. Nature (London) 2003.

-BALLANTYNE, Ford; KERKHOFF, A.J. The observed range for temporal mean-variance scaling exponents can be explained by reproductive correlation. Oikos 2007.

Reviewer 3 Report

I read the manuscript titled Taylor’s law in innovation processes and I consider that this work can be published without changes.

The model is presented correctly and the mathematical formalism is very good.

In my opinion the authors only should explain the criteria to choose the parameters ρ´s and ν in the simulations because, along the model, only those parameters are described but not the way to choose them.

Author Response

We thank you for your appreciation of our work and for your appropriate suggestion. We accordingly modified the captions of figures 1 and 5.